# How are brands associated by users in short videos—A study on the mechanism of user associations with brand placements in short videos based on signal theory

Zhang Yang[1,2], Sun Dongqi[3]*

1 Management Science and Engineering Research Centre, Jiangxi Normal University, Nanchang, Jiangxi, China, 2 Department of Economics and Trade, Tongling Polytechnic, Tongling, Anhui, China, 3 School of Business, Xinyang Vocational College of Art, Xinyang, Henan, China

* 1527425954@qq.com

**Data Availability Statement:** All relevant data are within the paper and its Supporting Information files.

## Abstract

The emergence of short video platforms has opened new avenues and opportunities for brand marketing. This paper investigates the mechanisms of brand dissemination in short videos, examining strategies for fostering brand associations to fulfill communication objectives. Drawing on signal theory, the study identifies the perceived value of short videos as the source of signals, with brand value and transparency serving as mediating factors, and brand association as the outcome. The research employs hypothesis testing and model building, supplemented by analysis of 560 valid questionnaires from research platforms, to scrutinize these mechanisms. Findings suggest that brand transparency and value effectively mediate the relationship between the perceived value and brand association, with brand transparency also mediating between perceived personality, utility, and brand association. Finally, this paper outlines management implications and acknowledges limitations based on the results.

## 1. Introduction

In the digital age, the paradigm of brand communication is evolving from traditional one-way dissemination toward more diverse and interactive approaches. The "2022 China Consumer Trend Report" suggests that brands should leverage multiple online communication channels to augment their influence and foster long-term trust with users, thereby building valuable user cognitive assets. Notably, short videos have emerged as a significant channel, experiencing rapid growth and attracting a vast user base. The "Statistical Report on the Development of the Internet in China" (released March 2, 2023) indicates that by the end of 2022, the number of Chinese short video users surpassed one billion, achieving a usage rate of 94.8% [1]. The varied publishing methods of short videos, coupled with the application of big data technology, have considerably enhanced user reach and provided a more personalized experience. Additionally, the allure of short videos has intensified due to creators' innovative content, making these

**Funding:** The updated submission includes revised funding information, highlighting the grants that provided both initial concept and financial support for this research. Specifically, the Outstanding Young Talent Project of the Anhui Provincial Department of Education in 2023 (YQYB2023154) and the Key Project of Humanities and Social Sciences of the Anhui Provincial Department of Education in 2022 (2022AH052746) served as the genesis for this research topic and were focal points of the project. Furthermore, the Anhui Province Philosophy and Social Science Key Laboratory Open Fund Project for Copper Industry Development Intelligent Decision-Making (2024tlxytyfzxm03) rendered financial support for materials and expenses during the revision process. Detailed amendments in the Funding section reflect support from these projects.

**Competing interests:** The authors have declared that no competing interests exist.

videos a crucial influence on user decisions and actions [2]. For instance, "Don't Sleep on Sleep," a short video by Mattress Firm—the largest mattress retailer in the United States—explores the effects of poor sleep on physical, mental, and emotional health using contextual content, directly boosting sales by 13.7%. Similarly, Starbucks' "Every Table Has a Story" illustrates that the cafe serves as a haven for comfort, inspiration, and community interaction, evidenced by high viewership rates across multiple platforms: 56% on Twitter, 91% on YouTube, and 89% on Facebook/Instagram. Nevertheless, not all brands achieve success through short videos [3]. Audi, for example, suffered reputational damage due to plagiarism in the script of "Moderately Content with Life" [4], and Yeshu Company faced a fine of 400,000 yuan along with criticism from the National Radio and Television Administration for controversial content in their short videos and live broadcasts [5]. These challenges underscore the complexities of effective brand communication through short videos, marking it as a critical area of focus.

Research on brand marketing within short videos remains nascent, primarily concentrating on three core areas: Initially, scholars analyze the attributes of short videos and their influence on brand communication effectiveness. For instance, YAO & KIM [6] examined authenticity dimensions from a user's perspective, which include originality, relevance, transparency, and experientiality, investigating their impact on both individual and corporate brand communication. Similarly, Ma et al. [7], studying clothing brands, discovered that the characteristics and presentation of short videos directly influence users' perceptions of brand quality Moreover, research suggests that the unique characteristics of users modulate their content receptivity and, consequently, the efficacy of brand marketing. Holmes (2021) explored the correlation between short video users' self-brand identification and brand communication effectiveness [8], while Yang et al. [9] demonstrated that users' self-congruence significantly impacts brand marketing effectiveness in their study of tourism destination brand ambassadors in short videos. Additionally, scholars have begun to examine the psychological traits of users during the information reception process as a vital component of their studies. Cho [10] deemed user experience a crucial factor in analyzing brand awareness in short videos. Aslan Oğuz et al. [11] further assessed how user engagement affects brand marketing effectiveness. These studies offer fresh insights into the integration of brand information within short videos and delve deeply into brand-related intentions and behaviors. However, the crucial initial step of brand recall by short video users for effective marketing has yet to be addressed in current research. Additionally, while it is evident that a brand's inherent characteristics impact its marketing success in short videos, this area remains unexplored.

This article, grounded in the framework of signal theory, examines effective strategies for disseminating brand messaging within short video environments and delineates a mechanism that enhances corporate brand marketing efficiency and effectiveness. The study addresses three principal questions: first, what elements in short videos enable users to perceive brand information; second, which brand features influence associations in these videos; and third, what mechanisms underlie these brand associations. Initially employing signal theory, the research investigates the perceived values users extract from short videos and their specific contributing factors. Data gathered through online surveys subsequently allows for an empirical analysis, which confirms the relationship between these factors and brand associations. Brand transparency and value are then examined as mediating variables, their roles verified to explore how they influence the formation of associations in short videos. Finally, informed by these findings, the paper offers actionable recommendations for enterprises on brand dissemination strategies in short videos.

The theoretical contribution of this paper lies in its exploration and validation of critical factors in the dissemination of brand information in short videos through signal theory: perceived values, brand features (transparency and value), and user brand associations. Moreover,

the study elucidates the mechanisms of these factors, thereby broadening the scope of research in brand marketing within short video contexts and extending the application reach of signal theory.

## 2. Theoretical background

Short videos typically consist of videos less than 10 minutes long, available on digital platforms such as the internet, platforms, and apps [12]. Signal theory divides business signals into three categories: signal sources, signals, and recipients [13, 14]. Within this framework, the short video itself is the signal source, the embedded brand information is the signal, and the viewer is the recipient. More precisely: Firstly, factors such as viewing duration, focus, and immersion significantly influence how users perceive brand information in short videos [15]. Enhancing these elements is crucial for increasing the perceived value of short videos [16]. Xiao et al. [17] employ signal theory and the U&G theory to claim that the perceived usefulness of short videos contributes to their value. Users feel greatly benefited when encountering content that piques their interest but lies outside their expertise. Algorithm-driven recommendations on short video platforms tailor content to user behavior and preferences, boosting personalized engagement and the perceived value of the short videos. Furthermore, users can voice their opinions through interactive features like bullet chats, comment sections, and private messages, thereby enhancing the content's perceived value [18]. Thus, this paper identifies three core components of the signal source in short videos: perceived usefulness, personalization, and interactivity. Secondly, in the realm of short videos, brand information serves as a signal, necessitating observable and valuable characteristics [13, 19]. These traits significantly affect brand consciousness among users [20]. For instance, brand transparency clarifies information about product quality, enhancing recognition of the brand's quality within short videos [21], and ensuring that the brand information is noticeable [22]. Additionally, the distinctiveness and differentiation conveyed by brand value help users identify and appreciate value differences among competing brands, enabled by digital tools on short video platforms [23]. Finally, as recipients of the signal, users have traits that profoundly impact the effectiveness of signal transmission [24]. User attention and comprehension levels [25, 26] are vital when content aligns well with user needs, resulting in heightened engagement and understanding, leading to an immersive experience [27]. In such cases, users not only notice but also understand and remember brand information through associative branding, thereby achieving effective brand communication [28].

## 3. Hypotheses development

### 3.1 Perceived value and brand association

Leveraging advancements in algorithmic technologies, short video platforms can now tailor content to align precisely with user preferences [29]. Such alignment ensures that users find the content highly suitable, which fosters interest and facilitates easier association with brand information during the viewing experience [30]. When short videos meet users' value expectations, they experience a sense of fulfillment, increasing their focus on the content and enhancing the association with brand information [31]. This heightened interest may lead users to actively participate by posting comments or engaging in discussions, thus stimulating interactions with other viewers. Such interactions further facilitate brand associations related to the content being discussed [29]. Collectively, these elements contribute to users' evaluation of the video's value, influencing their decision to continue watching or disengage [15]. Consequently, the perceived value of short videos plays a critical role in fostering brand associations among users.

Based on this premise, the following hypothesis is proposed:

H1: The perceived value in short videos positively influences users' brand associations;

H1a: Perceived personalization has a positive effect on users' brand associations;

H1b: Perceived usefulness positively impacts users' brand associations;

H1c: Perceived interactivity positively affects consumers' brand associations.

## 3.2 Perceived value and brand transparency

Short videos that cater to users' personalized needs can engage their interest, leading to an immersive experience with the content. During this immersion, users are likely to recognize or perceive authenticity in the short video information, including brand information, which positively influences brand transparency [32]. The content of short videos can broaden users' knowledge horizons, enhancing their perception of the video's utility. As users perceive utility, they tend to trust the video content, thereby fostering their recognition of brand transparency within the video [33]. Hence, users' perceived usefulness in short videos positively impacts brand transparency. Additionally, in the context of short videos, users interact with video creators and other users through bullet comments and the comment section. This interaction deepens their understanding of the video content, enabling them to independently evaluate the conveyed information, which may lead to a perception of brand transparency in the video [31]. Therefore, the perceived value of short videos can positively influence brand transparency.

In summary, this paper proposes the following hypothesis:

H2: The perceived value in short videos positively influences brand transparency;

H2a: Perceived personalization positively affects brand transparency;

H2b: Perceived usefulness positively impacts brand transparency;

H2c: Perceived interactivity positively influences brand transparency.

## 3.3 Perceived value and brand value

In the realm of short-form video, content tailored to users' personal needs significantly enhances their focus on the video content [34]. This heightened focus enables users to appreciate the value of both the video content and the embedded brand information [35], thereby positively influencing their perception of the brand's value. The utility of short video content is a direct determinant of consumers' value assessments [36]. Given that brand details are interwoven into the content, users instinctively form valuable perceptions and judgments. Consequently, the utilitarian aspect of short videos positively affects users' brand value perception. Interactive engagement in short videos deepens users' comprehension of the content, including the integrated brand information. Frequent interaction fosters trust in both the content and the brand, ultimately enhancing the brand value perceived by the users [37]. Thus, users' perceived value from short videos can beneficially impact the overall brand value.

In summary, this paper proposes the following hypothesis:

H3: The perceived value in short videos has a positive impact on brand value;

H3a: Perceived personalization positively influences brand value;

H3b: Perceived usefulness positively influences brand value;

H3c: Perceived interactivity positively influences brand value.

## 3.4 Brand transparency and brand association

In the context of short videos, a brand's position in the minds of users directly influences their associations and perceptions of that brand. The recognition of a brand's quality and its positioning significantly impacts these associations. Brand transparency relates to user perceptions regarding the quality and authenticity of a brand. When users perceive a high level of brand transparency in short videos, they immediately focus on the brand upon its appearance, eliciting a series of associations such as the brand name and image [38]. Consequently, the following hypothesis is proposed:

H4: In the context of short videos, brand transparency has a positive impact on brand associations.

## 3.5 Brand value and brand association

Brand value significantly influences user attention towards a brand. When users perceive a brand as highly valuable, their level of attention increases accordingly, and vice versa [39]. Within the context of short videos, brands perceived as valuable are more likely to capture user attention [40]. When users are deeply engaged with a short video and the embedded brand holds high perceived value, it will prompt the formation of brand associations [41]. Consequently, the following hypothesis is proposed:

H5: In the context of short videos, brand value has a positive impact on brand associations.

## 3.6 The mediating role of brand transparency and brand value

While watching short videos, users experience a "sense of immersion" attributed to the video's inherent value. This deep immersion in video content often leads to profound brand associations stimulated by the brand's unique features [42]. Brand transparency refers to users' perceptions of a brand's quality and authenticity. When established, these perceptions increase the likelihood of triggering brand associations during the presentation of the brand in short videos [43]. Similarly, brand value—users' assessment of a brand's worth—when perceived as high, enhances the depth of brand associations within short videos [41]. Consequently, the study posits the following hypotheses:

H6: Brand transparency serves as a mediator between perceived value and brand associations.

H6a: Brand transparency mediates the relationship between perceived personality and brand associations.

H6b: It also mediates between perceived usefulness and brand associations.

H6c: Lastly, brand transparency bridges perceived interactivity and brand associations.

H7 proposes that brand value acts as a mediator between perceived value and brand associations, with H7a, H7b, and H7c further delineating this mediation with respect to perceived personality, practicality, and interactivity, respectively.

## 3.7 Research model

Based on the hypotheses above, the paper constructed the following model:

## 4. Methodology

### 4.1 Measurement development

According to the research model presented in this article, the variables examined include perceived personality, perceived usefulness, perceived interactivity, brand transparency, brand value, and brand associations. This article synthesizes the findings from relevant literature. The variable for perceived personality incorporates four measurement items proposed by Zhang et al. [48]; perceived usefulness is informed by studies from DUCOFFE [49] and Yan et al. [12], applying four measurement items; perceived interactivity uses four measurement items, combining research from Heinonen [50] and Yan et al. [12]; brand transparency includes five measurement items, drawing on studies by Yang and Battocchio [44] and Konuk [21]; brand value integrates five measurement items based on research from Aaker [45] and Baumgarth & Schmidt [46]; and brand associations, with five measurement items, emanate from the study by Yoo and Donthu [47]. Further details can be found in Table 1.

### 4.2 Questionnaire design

This article employs the described measurement items as integral components of the questionnaire, integrating short video scenarios with a 5-point Likert scale for evaluation—where 5 indicates "strongly agree," 4 "somewhat agree," 3 "neutral," 2 "somewhat disagree," and 1 "strongly disagree." Additionally, the article includes a section on basic demographic information to align the characteristics of the survey participants with the research objectives. This section collects data on gender and age distribution, experience with short videos, and brand familiarity. From this information, researchers can ascertain whether participants possess sufficient familiarity with short videos and brands to qualify as subjects for this study.

Upon drafting the initial survey questionnaire, the authors sought to enhance both its professionalism and relevance. They engaged peer reviewers to critique the questionnaire, implementing adjustments based on this feedback. Furthermore, to ensure comprehensive understanding of the survey questions among respondents, a preliminary survey was conducted. Comments from this early feedback prompted refinements in the question wording, leading to the finalization of the official research questionnaire.

### 4.3 Sample and study case

The China Online Audio-visual Development Report (2024) reveals that by December 2023, the number of online audio-visual users in China reached 1.074 billion, with a total of 1.55 billion short video accounts across the internet [51]. As a result, there is substantial usage and deep market penetration of short videos among Chinese users [52]. Considering China's vast population engaged in using short videos, alongside rapid economic growth and the pervasive availability of smart devices, the conditions for usage are favorable and experiences are extensive. Thus, this demographic has been selected as the sample for this study. To guarantee the randomness of the research, surveys will be collected randomly using the paid questionnaire platform Credamo, with participants drawn from a variety of Chinese provinces, ensuring a randomized sample distribution.

### 4.4 Data collection

This study utilized Credamo, a leading paid survey platform in China known for its comprehensive services including questionnaire design, access to millions of online participants, and advanced visual statistical modeling. Credamo, an acronym for Creator of Data and Model (https://www.credamo.com/), serves over 3,000 universities worldwide and 4,000 companies,

**Table 1. Measurement of variables.**

| Variables | Items | | Source |
|---|---|---|---|
| Brand Transparency (tran) | tran1 When the brand's name, logo, and other basic information are very clear in the short video, I will notice the brand in the short video. | | (Yang and Battocchio; Konuk), [21, 44] |
| | tran2 When the content promoted by the brand in the short video is credible, I will notice the brand in the short video. | | |
| | tran3 When the brand clearly displays its positioning and main business in the short video, I will notice the brand in the short video. | | |
| | tran4 When I feel that embedding brand advertisements in the short video context is appropriate and there is no sense of dissonance, I will notice the brand in the short video. | | |
| | tran5 When the brand content embedded in the short video is consistent with what I previously knew about the brand, I will notice the brand in the short video. | | |
| Brand Value (bv) | bv1 When I feel that the brand's product quality is better than others, I will notice the brand in the short video. | | Aaker; Baumgarth and Schmidt, [45, 46] |
| | bv2 When I think the brand is more well-known than others, I will notice the brand in the short video. | | |
| | bv3 When I frequently use the brand's products, I will notice the brand in the short video. | | |
| | bv4 When I believe the brand's products offer better value for money, I will notice the brand in the short video. | | |
| | bv5 When the brand's products become my first choice while shopping, I will notice the brand in the short video. | | |
| Brand Association (ba) | ba1 By watching short videos, I can quickly recall some features of the brands that appear in the video. | | Yoo and Donthu, [47] |
| | ba2 By watching short videos, I can quickly recognize the names or logos of the brands that appear in the video. | | |
| | ba3 By watching short videos, I can quickly think of the brand's advertising slogans, spokespersons, brand stories, etc. | | |
| | ba4 Watching short videos makes my memory of the brand more profound. | | |
| | ba5 While watching short videos, when I see information about the brand, it brings the brand to mind. | | |
| Perceived Value in Short Form Videos | Perceived Personality (pp) | pp1 I feel that every short video I watch is able to attract me. | Zhang et al. [48] |
| | | pp2 I feel that the short videos I see know what content I want to watch. | |
| | | pp3 I feel that the short videos I watch meet my viewing needs. | |
| | | pp4 I think the short video platform already knows my viewing preferences and recommends similar content based on those preferences. | |
| | Perceived Usefulness (pu) | pu1 I feel that I can learn quite a bit of knowledge from short videos. | DUCOFFE; Yan et al. [12, 49] |
| | | pu2 I feel that I can obtain quite a bit of valuable information from short videos. | |
| | | pu3 I believe the knowledge learned from short videos can help me make decisions. | |
| | | pu4 I think short videos can provide me with useful and timely information. | |
| | Perceived Interactivity (pi) | pi1 I believe I can communicate with others through comments in short videos. | Heinonen; Yan et al. [12, 50] |
| | | pi2 I feel that I can communicate smoothly with others in the comment section of short videos. | |
| | | pi3 I think short videos provide many ways that facilitate communication with others. | |
| | | pi4 I believe when I comment or post comments in short videos, I receive timely responses from the video creators or others. | |

accommodating over 3 million online participants. Its robust dataset facilitates the identification of participants whose profiles align precisely with specific questionnaire criteria, enhancing both the accuracy of the participant selection and the overall quality of research data.

To reduce potential data inaccuracies resulting from concentrated survey responses within a brief timeframe, the study was structured in two phases. The initial phase commenced in December 2023, securing 256 completed questionnaires, followed by the second phase in February 2024, which gathered 344 questionnaires. From a total of 600 distributed, questionnaires with completion times under 100 seconds, repetition rates exceeding two-thirds, or more than

one-third blank responses were disqualified, leaving 560 valid questionnaires. This resulted in a validity rate of 93.3%.

## 5. Data analysis and results

### 5.1 Analysis of characteristics of research subjects

Table 2 reveals a higher proportion of female participants, consistent with observations that females tend to use mobile phones more frequently and for longer periods compared to males. The participants are predominantly aged 18–35, representing 83.57% of the sample, a demographic recognized for significant mobile usage skills and experience. The most popular short video platforms among the participants include Tiktok (91.40%), Xiaohongshu (64.3%),

**Table 2. Characteristics of research subjects.**

| Question | items | Proportion |
|---|---|---|
| Gender | A、Male | 35.71% |
| | B、Female | 64.29% |
| Age(years old) | A.0-18 | 0.00% |
| | B.18-25 | 30.71% |
| | C.26-35 | 52.86% |
| | D.36-45 | 13.57% |
| | E.45 -55 | 2.14% |
| | F.Over 55 | 0.71% |
| What platforms do you often use to watch short videos? | A.Tiktok | 91.40% |
| | B.Kuaishou | 54.30% |
| | C.Xiaohongshu | 64.30% |
| | D.WeChat | 49.30% |
| | E.Xigua | 10.70% |
| | F.Meipai | 5.00% |
| | G.QQ Short video | 6.40% |
| | H.The Short video in internet explore(such as UC, Firefox and other big fish video) | 8.60% |
| | I.The video app like IQIYI and Tencent | 40.70% |
| | J. Sohu, NetEase and other news platform APP | 12.10% |
| | L.Others | 7.90% |
| Please think about how much time you spend on short videos each day? | A、0–20 minutes | 1.43% |
| | B、21–40 minutes | 8.57% |
| | C、41–60 minutes | 17.14% |
| | D、1–2 hours | 37.86% |
| | E、2–4个hours | 27.14% |
| | F、over 4 hours | 7.86% |
| Which of the following descriptions of brand familiarity fits your situation? | A.No matter what field, I am familiar with the products of the brand | 3.57% |
| | B. In most areas, the brand is familiar | 35.00% |
| | C. In some specific areas, the brand is still familiar | 27.86% |
| | D.I only care about the brand in my favorite field, other I am not familiar with | 22.14% |
| | E.I am only familiar with a few brands, and I am not familiar with other brands in the field | 11.43% |
| | F.I am not familiar with the brand at all, just recognize the logo | 0.00% |

Kuaishou (52.86%), WeChat Short Video (49.3%), and drama-series platforms such as iQIYI and Tencent Video (40.7%). These platforms constitute the mainstream video platforms in China, corroborating the data with real-world patterns. The majority of respondents spend 1–2 hours (37.86%) and 2–4 hours (27.14%) daily on these platforms, evidencing their extensive experience with short videos. Concerning brand familiarity, 35% of participants are familiar with brands across most sectors, 27.86% in specific sectors, and 22.14% only within their interests, indicating a high level of brand awareness, with no participant completely unfamiliar with brands.

The demographic and behavioral profiles of the participants align with the typical user base of short video platforms, as indicated by their substantial experience and notable brand familiarity. This alignment supports the research's thematic needs, affirming the appropriateness of the participant selection.

## 5.2 Descriptive analysis of questionnaire items

Table 3 indicates that the mean values of the survey items are between 3 and 4, and all standard deviations are below 0.1. Furthermore, the skewness and kurtosis values are under 3, confirming that the study's data adhere to a normal distribution.

**Table 3. Descriptive analysis.**

| items | M-value | St-error | Skewness | | Kurtosis | |
|---|---|---|---|---|---|---|
| pp1 | 3.99 | 0.03 | (0.62) | 0.10 | 0.84 | 0.21 |
| pp2 | 4.21 | 0.04 | (1.11) | 0.10 | 1.19 | 0.21 |
| pp3 | 4.18 | 0.03 | (0.51) | 0.10 | (0.34) | 0.21 |
| pp4 | 4.43 | 0.03 | (1.47) | 0.10 | 4.08 | 0.21 |
| pu1 | 4.08 | 0.04 | (0.96) | 0.10 | 0.84 | 0.21 |
| pu2 | 4.19 | 0.03 | (1.12) | 0.10 | 1.50 | 0.21 |
| pu3 | 3.81 | 0.04 | (0.79) | 0.10 | 0.29 | 0.21 |
| pu4 | 4.22 | 0.03 | (0.81) | 0.10 | 0.73 | 0.21 |
| pi1 | 3.84 | 0.05 | (0.71) | 0.10 | (0.54) | 0.21 |
| pi2 | 3.99 | 0.04 | (1.14) | 0.10 | 0.88 | 0.21 |
| pi3 | 3.98 | 0.04 | (1.20) | 0.10 | 1.97 | 0.21 |
| pi4 | 3.81 | 0.04 | (0.71) | 0.10 | (0.29) | 0.21 |
| tran1 | 4.16 | 0.03 | (0.76) | 0.10 | 0.36 | 0.21 |
| tran2 | 4.26 | 0.03 | (1.10) | 0.10 | 1.45 | 0.21 |
| tran3 | 4.12 | 0.03 | (0.80) | 0.10 | 1.25 | 0.21 |
| tran4 | 4.17 | 0.04 | (1.08) | 0.10 | 0.90 | 0.21 |
| tran5 | 4.28 | 0.03 | (1.31) | 0.10 | 2.72 | 0.21 |
| bv1 | 4.36 | 0.03 | (0.90) | 0.10 | 0.39 | 0.21 |
| bv2 | 4.26 | 0.03 | (0.77) | 0.10 | 0.35 | 0.21 |
| bv3 | 4.34 | 0.03 | (0.53) | 0.10 | 0.42 | 0.21 |
| bv4 | 4.27 | 0.03 | (0.80) | 0.10 | 0.45 | 0.21 |
| bv5 | 4.46 | 0.03 | (1.12) | 0.10 | 1.33 | 0.21 |
| ba1 | 3.99 | 0.04 | (0.56) | 0.10 | (0.23) | 0.21 |
| ba2 | 4.16 | 0.04 | (1.03) | 0.10 | 0.69 | 0.21 |
| ba3 | 3.76 | 0.04 | (0.87) | 0.10 | 0.35 | 0.21 |
| ba4 | 4.32 | 0.03 | (0.78) | 0.10 | 0.44 | 0.21 |
| ba5 | 4.28 | 0.03 | (0.94) | 0.10 | 1.07 | 0.21 |

Pp,Perceived Personality;pu,Perceived Usefulness;pi,Perceived Interactivity;tran,Brand Transparency;bv,Brand Value;ba,Brand Association.

### 5.3 Test for common method bias

This research utilizes the Harman single-factor test to evaluate common method bias. Analysis reveals six factors with eigenvalues exceeding 1, with the principal factor explaining 32.424% of the variance—below the critical 40% threshold. These findings indicate an absence of significant common method bias.

### 5.4 Validity and reliability

In terms of reliability, the Cronbach's Alpha values for the independent, mediator, and dependent variables were 0.85, 0.86, and 0.74 respectively, all surpassing the threshold of 0.6. Furthermore, corrected item-total correlations should exceed 0.4. With the exception of item pp4, which recorded a value of 0.25 and thus did not meet this threshold, all other items were compliant; consequently, item pp4 was excluded from subsequent analysis.

For the validity analysis (Table 4), the Kaiser-Meyer-Olkin (KMO) measure was computed along with an exploratory factor analysis to assess the alignment between the items and the predefined dimensions. The results indicated that all KMO values exceeded 0.78. Except for item trans5, which achieved a factor loading of 0.49, all other items exhibited factor loadings above 0.5, leading to the exclusion of item trans5. Additionally, the cumulative explanatory variance percentages for the independent, mediator, and dependent variables significantly surpassed 50%. The exploratory factor analysis confirmed that all items were consistent with their respective presets, demonstrating strong construct validity of the scale. All variables' Average Variance Extracted (AVE) values met the acceptable criterion (AVE > 0.36), and the Composite Reliability (CR) scores were all above 0.7, confirming robust convergent validity for each variable.

In assessing discriminant validity (Table 5), this paper initially centralizes the aforementioned six variables. Post-centralization, the processed data for each variable adequately represents the respective items. Correlation analyses are conducted using SPSS, where discriminant validity is evaluated based on the squared AVE values. Referring to the subsequent table, Pearson correlation coefficients are uniformly below 0.01, signifying substantial correlations among variables. The diagonal displays the square roots of the AVE values, which exceed the inter-variable correlations, thus confirming the robust discriminant validity of the variables presented in this study.

## 6. Hypotheses testing

This study employs AMOS for evaluating the research model. Considering that the hypotheses comprise both primary and secondary hypotheses, the evaluation is structured in two phases: the initial phase assesses the primary hypotheses, followed by the sub-hypotheses in the subsequent phase.

### 6.1 Primary hypotheses testing

This study utilizes centralized data on Perceived Value (PV), Brand Transparency, Brand Value, and Brand Associations to develop a manifest variable model within AMOS, and applies the Bootstrap method to examine mediating effects. Detailed results are presented below:

Firstly, Table 6 shows that in the context of short videos, perceived value significantly affects brand transparency, brand value, and brand associations. Moreover, brand transparency and value positively influence brand associations, thereby confirming Hypotheses H1 through H5. The impact of perceived value on brand transparency is notably the most substantial,

**Table 4. Analysis of validity and reliability.**

| Constructs | Items | Loading | | | | Corrected item-total correlation | Cronbach's Alpha value after item deletion | AVE | CR |
|---|---|---|---|---|---|---|---|---|---|
| Independent Variable:Cronbach's Alpha = 0.85; KMO值 = 0.83, sig = 000; Cumulative Explained Total Variance = 63.70% | | | | | | | | | |
| Perceived Value in Short Form Videos | pp1 | 0.78 (0.67) | | | | 0.50 | 0.84 | 0.48 | 0.73 |
| | pp2 | 0.50 (0.55) | | | | 0.48 | 0.84 | | |
| | pp3 | 0.77 (0.62) | | | | 0.44 | 0.85 | | |
| | pp4 | (0.78) | | | | 0.25 | 0.86 | | |
| | pu1 | | 0.79 | | | 0.51 | 0.84 | 0.61 | 0.86 |
| | pu2 | | 0.77 | | | 0.52 | 0.84 | | |
| | pu3 | | 0.80 | | | 0.46 | 0.84 | | |
| | pu4 | | 0.75 | | | 0.55 | 0.84 | | |
| | pi1 | | | 0.86 | | 0.58 | 0.84 | 0.66 | 0.89 |
| | pi2 | | | 0.79 | | 0.69 | 0.83 | | |
| | pi3 | | | 0.75 | | 0.60 | 0.83 | | |
| | pi4 | | | 0.84 | | 0.67 | 0.83 | | |
| Mediating Variable: Cronbach's Alpha = 0.86; KMO Value = 0.85,sig = 000; Cumulative Explained Total Variance = 50.72% | | | | | | | | | |
| Brand Transparency | tran1 | | | 0.78 (0.79) | | 0.54 | 0.81 | 0.46 | 0.77 |
| | tran2 | | | 0.63 (0.62) | | 0.52 | 0.81 | | |
| | tran3 | | | 0.52 (0.51) | | 0.41 | 0.82 | | |
| | tran4 | | | 0.74 (0.73) | | 0.42 | 0.82 | | |
| | tran5 | | | (0.49) | | 0.57 | 0.80 | | |
| Brand Value | bv1 | | | | 0.56 | 0.62 | 0.80 | 0.46 | 0.81 |
| | bv2 | | | | 0.71 | 0.56 | 0.80 | | |
| | bv3 | | | | 0.66 | 0.48 | 0.81 | | |
| | bv4 | | | | 0.71 | 0.58 | 0.80 | | |
| | bv5 | | | | 0.72 | 0.46 | 0.81 | | |
| Dependent Variable: Cronbach's Alpha = 0.74; KMO Value = 0.78,sig = 0.000; Cumulative Explained Total Variance = 49.74% | | | | | | | | | |
| Brand Association | ba1 | | | | 0.74 | 0.55 | 0.68 | 0.50 | 0.83 |
| | ba2 | | | | 0.76 | 0.57 | 0.67 | | |
| | ba3 | | | | 0.69 | 0.49 | 0.71 | | |
| | ba4 | | | | 0.62 | 0.42 | 0.72 | | |
| | ba5 | | | | 0.71 | 0.51 | 0.70 | | |

Pp,Perceived Personality;pu,Perceived Usefulness;pi,Perceived Interactivity;tran,Brand Transparency;bv,Brand Value;ba,Brand Association.

indicating that users who highly value short videos also perceive greater transparency in the featured brands. Additionally, brand transparency is the most influential factor affecting brand associations; a higher perception of brand transparency significantly enhances brand recall.

According to the Sobel mediation test procedure, these findings imply that brand transparency and brand value might mediate the effects of perceived value on brand associations. This paper employs the Bootstrap method in AMOS to verify the mediation effect, which is explained in the subsequent section.

**Table 5. Analysis of discriminant validity.**

|  | Pp centralization | Pu centralization | Pi centralization | Bv centralization | Tran centralization | Ba centralization |
|---|---|---|---|---|---|---|
| Pp centralization | 0.69 |  |  |  |  |  |
| Pu centralization | 0.40** | 0.81 |  |  |  |  |
| Pi centralization | 0.51** | 0.39** | 0.78 |  |  |  |
| Bv centralization | 0.33** | 0.33** | 0.52** | 0.68 |  |  |
| Tran centralization | 0.53** | 0.49** | 0.62** | 0.60** | 0.68 |  |
| Ba centralization | 0.50** | 0.51** | 0.53** | 0.53** | 0.67* | 0.71 |

Pp,Perceived Personality;pu,Perceived Usefulness;pi,Perceived Interactivity;tran,Brand Transparency;bv,Brand Value;ba,Brand Association.

Table 7 clearly demonstrates that the indirect effects remain above zero within the 95% confidence interval, suggesting that brand attributes, namely brand transparency and brand value, serve as mediators for perceived value and brand associations in short video contexts, thereby supporting hypotheses H6 and H7. Moreover, in these contexts, brand transparency contributes to 46% of the indirect effects, compared to only 10% from brand value, highlighting a more pronounced mediating influence of brand transparency.

## 6.2 Sub-hypotheses testing

As mentioned earlier, this paper centralizes perceived personality (pp), perceived usefulness (pu), and perceived interactivity (pi), constructs a manifest variable model in AMOS, and employs the Bootstrap procedure to test for mediating effects, as detailed below:

According to Table 8, in the context of short videos, perceived personality and perceived usefulness both significantly and positively influence brand association; hence, hypotheses H1a and H1b are supported. However, perceived interactivity does not significantly impact brand association (p = 0.06), thus H1c is not supported. Notably, perceived personality, perceived usefulness, and perceived interactivity all contribute positively to brand transparency, supporting H2a, H2b, and H2c. Furthermore, while perceived usefulness and perceived interactivity have a significant positive effect on brand value, perceived personality does not, thereby validating H3b and H3c but not H3a. Additionally, both brand transparency and brand value significantly enhance brand association, supporting H4 and H5. Specifically in short videos, perceived usefulness has the most substantial effect on brand association, with users likely forming associations when they recognize practicality in these videos. Moreover, perceived interactivity most significantly affects brand transparency, enhancing user trust and the perception of transparency. Similarly, perceived interactivity also profoundly influences

**Table 6. Standardized path coefficients and significance of main assumptions.**

|  | Standardized Estimate | S.E. | C.R. | P |
|---|---|---|---|---|
| tran<—PV | 0.66 | 0.01 | 20.55 | *** |
| bv<—PV | 0.52 | 0.02 | 14.38 | *** |
| ba<—tran | 0.38 | 0.05 | 10.04 | *** |
| ba<—bv | 0.17 | 0.04 | 5.10 | *** |
| ba<—PV | 0.32 | 0.02 | 7.62 | *** |

Pp,Perceived Personality;pu,Perceived Usefulness;pi,Perceived Interactivity;tran,Brand Transparency;bv,Brand Value;ba,Brand Association.

Table 7. Primary hypothesis mediation effect test.

| Mediator variable | Indirect effect | Total effect | Proportion of indirect effect | 95% confidence interval of the indirect effect | |
|---|---|---|---|---|---|
| | | | | LICI | ULCI |
| Brand Transparency | 0.12 | 0.30 | 40% | 0.09 | 0.15 |
| Brand Value | 0.04 | 0.30 | 13% | 0.03 | 0.06 |

brand value, suggesting that effective interactivity in short videos enhances the perceived value of featured brands.

Regarding the Sobel mediation test procedure, since perceived interactivity does not significantly influence brand association, neither brand transparency nor brand value mediates between perceived interactivity and brand association, invalidating H6c and H7c. Similarly, the non-significant impact of perceived personality on brand value means that brand value does not mediate between perceived personality and brand association, leading to the rejection of hypothesis H7a. The analysis of other variables requires a Bootstrap test, with results presented subsequently.

From the results in Table 9, it is evident that, on one hand, the indirect effects did not cross zero within the 95% confidence interval, hence hypotheses H6a, H6b, and H7b are supported; on the other hand, in the context of short videos, the proportion of indirect effects involving perceived usefulness and brand association is higher for brand transparency than for brand value, indicating that brand transparency plays a stronger mediating role than brand value in the context of perceived usefulness.

## 7. Discussion

Studies on brand association reveal that users' perceived value from digital content fosters positive brand associations [53]. Specifically, short videos, predominantly composed of digital content, create deeply immersive scenarios [54]. Computing algorithms that recommend relevant content to users can significantly enhance brand association [55]. When users encounter content they consider practical in short videos, their increased focus often leads to recognition of, and association with, the featured brands [56]. However, contrary to prior findings where

Table 8. Sub-hypothesis standardized path coefficient and significance.

| | | Standardized Estimate | S.E. | C.R. | P |
|---|---|---|---|---|---|
| tran<—pp | | 0.22 | 0.04 | 6.10 | *** |
| tran<—pu | | 0.25 | 0.03 | 7.04 | *** |
| tran<—pi | | 0.44 | 0.02 | 12.59 | *** |
| bv<—pp | | 0.05 | 0.05 | 1.33 | 0.18 |
| bv<—pu | | 0.15 | 0.03 | 3.97 | *** |
| bv<—pi | | 0.46 | 0.03 | 12.33 | *** |
| ba<—tran | | 0.39 | 0.05 | 10.12 | *** |
| ba<—pu | | 0.22 | 0.03 | 6.39 | *** |
| ba<—pp | | 0.16 | 0.05 | 4.97 | *** |
| ba<—bv | | 0.21 | 0.04 | 5.65 | *** |
| ba<—pi | | 0.07 | 0.03 | 1.75 | 0.06 |

Pp,Perceived Personality;pu,Perceived Usefulness;pi,Perceived Interactivity;tran,Brand Transparency;bv,Brand Value;ba,Brand Association.

**Table 9. Sub-hypotheses mediation effect test.**

| Mediator variable | Indirect effect | Total effect | Proportion of indirect effect | 95% confidence interval of the indirect effect | |
|---|---|---|---|---|---|
| | | | | LICI | ULCI |
| Brand Transparency | 0.13 | 0.33 | 39.39% | 0.07 | 0.21 |
| Brand Transparency | 0.09 | 0.32 | 28.13% | 0.06 | 0.14 |
| Brand Value | 0.03 | 0.32 | 9.38% | 0.01 | 0.05 |

user interaction in various contexts positively impacts brand perception [57–59], interactions in short video settings, primarily via bullet comments or comment sections, can distract users. This distraction decreases their attention to embedded brands, thus minimally influencing brand association through perceived interactivity in these studies. Additionally, this research corroborates the critical roles of brand value and transparency; perceptions of brand transparency significantly influence users' views on authenticity and brand image in marketing [60], while perceived brand value impacts the brand's stature in the user's mind [61]. Enhanced within short video scenarios, both factors contribute positively to brand association [27].

In the context of short videos, brand transparency garners user attraction and contributes to their perception of the video's value [52]. When captivated by such content, users can recognize brands based on their transparency, thereby confirming the findings of this study. Short videos foster user focus through features like personalized recommendations, utilitarian value, and interactivity, all of which enhance the perception of brand transparency [53]. This research supports these observations. Regarding the mediating role of brand transparency, it has a limited impact on the relationship between perceived interactivity and brand association due to the minimal influence of perceived interactivity; however, other mediating effects have been endorsed.

Regarding brand value, users associate valuable brands with significant awareness and recognition [62]. In short videos, focused engagement with the content leads to an enhanced recognition of brands considered valuable by the users [7]. As attention toward short videos heightens, the likelihood of noticing embedded brand values increases [63]. This study corroborates these insights. Nonetheless, it was found that despite the substantial roles of perceived interactivity (0.45) and utility (0.15), perceived personalization—determined by platform algorithms [64]—does not significantly contribute to the perception of brand value. Concerning brand value's mediating role, tests indicate that the negligible effects of perceived interactivity and personalization do not support its mediating influence, yet brand value in contextually valuable short videos, particularly those with utilitarian content, can still foster brand associations.

## 8. Conclusion

This article, grounded in signal theory, delineates perceived value factors in short videos: three signal sources—perceived personality, utility, and interactivity; and two signal factors—brand transparency and value, along with a receiver factor, user brand association.

The study reveals that values perceived from short videos foster associations with embedded brands. Brand transparency and value notably mediate the relationship between perceived user value and brand association. Users' perceptions of brand transparency and value significantly influence their ability to recall brands seen in short videos. While short videos meet users' personalized and practical needs, enhancing brand associations, interactivity within the videos does not promote such associations. Interestingly, when users obtain practical

knowledge from short videos, their attentiveness increases, facilitating brand recall. However, interactions with creators or other viewers via comments or bullet chats shift attention away from brands embedded in the content. Users able to thoroughly comprehend a brand through short videos, predicated on practical utility, swiftly recall the brand, highlighting a mediating role of brand transparency between practicality and brand association. Moreover, users viewing videos for practical reasons are more focused and likely to associate with high-value brands within the videos, situating brand value as a mediator between perceived utility and brand association. Short videos tailored to user needs prompt quick brand recall when users fully understand the brand, underscoring the mediating function of brand transparency between perceived personality and brand association. Despite frequent interactions fostering some brand understanding and value recognition, low prior brand awareness prevents association formation, emphasizing the need for brands to improve their visibility.

## 9. Practical implication

Based on the findings discussed above, this paper offers several management recommendations aimed at promoting brand associations among users exposed to brands featured in short videos and increasing the effectiveness of brand communication:

First, to foster brand associations, brands should craft and select high-value short videos and incorporate brand-related information. Users generally pay more attention to high-quality short videos, thereby facilitating easier association with the embedded brand information. Thus, in the production and selection of short videos, it is imperative for brands to enhance content quality to bolster brand communication.

Second, brands should mitigate controversy in short video content and reduce direct user interactions. The study indicates that interactions do not foster brand associations, and they may even be detrimental. Due to the direct connection between controversial content and user interaction, minimizing such controversies can significantly enhance the effectiveness of brand communication. Furthermore, limiting user interactions lets brands focus their limited marketing resources more effectively.

Third, emphasizing brand transparency during the brand-building and marketing phase is crucial. Improving transparency in short video scenarios may help users easily detect brand information, leading to brand associations. This encompasses defining clear brand positioning at the early stages of brand development and conveying sincerity and authenticity throughout the marketing communications, which boosts transparency.

Fourth, brands should elevate the professional value of their positioning by embedding or creating highly functional video content to advance brand communication. As brand transparency and value mediate the perceived utility of short video content and brand association, enhancing functional characteristics relevant to the brand's positioning in short videos can increase both practicality and transparency, thus, fulfilling brand communication objectives in these contexts.

Fifth, it is vital for brands to establish partnerships with short video platforms to enable personalized recommendations of videos embedded with brand information tailored to the target demographic. Given the influence of brand transparency on perceived individuality and brand associations, it is recommended that brands, assuming established transparency, improve the level of personalized recommendations to facilitate the development of brand associations. Therefore, brands should collaborate with short video platforms to tailor recommendations of brand-embedded videos to their target audience, thus fostering brand associations and boosting communication efficacy.

## 10. Limitation and future research

This study is subject to several limitations:

Firstly, there is a diversity in user needs and corresponding value derivations from short videos. The factors of perceived value identified in this research are not exhaustive; societal shifts may necessitate the identification of additional factors, an endeavor currently constrained by research capabilities. This paper introduces variables such as perceived value, brand transparency, brand value, and brand association, refined under the auspices of signal theory. Nevertheless, numerous factors influencing the effectiveness of brand communication have yet to be explored and will require extensive future investigation by scholars.

Secondly, the study sample is relatively small. While the sample utilized aligns well with the required demographics and the data are reliable and valid, its coverage and volume are inadequate. Subsequent research should aim to enlarge the sample size and expand the population studied.

Thirdly, although this research constructs a mechanism for generating brand associations from short videos, it overlooks the dynamic nature of both mediating and potential moderating factors. The current model only considers brand transparency and brand value as mediating variables, excluding other potential factors. Future studies should investigate the mediating roles of brand characteristics more deeply and identify moderating variables to refine the model.

Finally, the depth of the research needs expansion. While it demonstrates that users associate brands with short videos, translating these associations into purchase intentions and behaviors requires a deeper understanding of how to imbue short videos with more profound emotional and functional values.

## Supporting information

**S1 Data.**
(XLSX)

## Author Contributions

**Writing – original draft:** Zhang Yang.

**Writing – review & editing:** Sun Dongqi.

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
