## [Decision Letter · Decision Letter 0]

15 Jul 2024

PONE-D-24-22415How are brands associated by users in short videos - A study on the mechanism of user associations with brand placements in short videos based on signal theoryPLOS ONE

Dear Dr. Sun,

Thank you for submitting your manuscript to PLOS ONE. After careful consideration, we feel that it has merit but does not fully meet PLOS ONE’s publication criteria as it currently stands. Therefore, we invite you to submit a revised version of the manuscript that addresses the points raised during the review process.

We look forward to receiving your revised manuscript.

Kind regards,

Hua Pang

Academic Editor

PLOS ONE

2. You indicated that ethical approval was not necessary for your study. We understand that the framework for ethical oversight requirements for studies of this type may differ depending on the setting and we would appreciate some further clarification regarding your research. Could you please provide further details on why your study is exempt from the need for approval and confirmation from your institutional review board or research ethics committee (e.g., in the form of a letter or email correspondence) that ethics review was not necessary for this study? Please include a copy of the correspondence as an ""Other"" file.

 [The Outstanding Young Talent Project of Anhui Provincial Department of Education in 2023 (YQYB2023154); and the Key Project of Humanities and Social Sciences of Anhui Provincial Department of Education in 2022 (2022AH052746) provided support for this study.].  

[This paper was supported by the Outstanding Young Talent Project of Anhui Provincial Department of Education in 2023 (YQYB2023154) and the Key Project of Humanities and Social Sciences of Anhui Provincial Department of Education in 2022 (2022AH052746)]

 [The Outstanding Young Talent Project of Anhui Provincial Department of Education in 2023 (YQYB2023154); and the Key Project of Humanities and Social Sciences of Anhui Provincial Department of Education in 2022 (2022AH052746) provided support for this study.]

6. We note that your Data Availability Statement is currently as follows: [All relevant data are within the manuscript and its Supporting Information files.]

Reviewers' comments:

Reviewer's Responses to Questions

**Comments to the Author**

1. Is the manuscript technically sound, and do the data support the conclusions?

Reviewer #1: Yes

Reviewer #2: Yes

2. Has the statistical analysis been performed appropriately and rigorously? 

Reviewer #1: Yes

Reviewer #2: Yes

3. Have the authors made all data underlying the findings in their manuscript fully available?

Reviewer #1: Yes

Reviewer #2: Yes

4. Is the manuscript presented in an intelligible fashion and written in standard English?

Reviewer #1: Yes

Reviewer #2: Yes

5. Review Comments to the Author

Reviewer #1: The paper has substantial contributions. The authors provided sufficient literature support, appropriate methodology including data collection and analysis methods. The findings are also acceptable. However, they can improve language and sentence structure.

Reviewer #2: The manuscript is of good quality and topic, but:

In the introduction, no specific scenario is followed. The importance of the subject? Target? Knowledge gap? Innovation? Theoretical and practical contribution?

In the literature, it has been tried to make an essay rather than analyzing the literature and examining the relationships between variables analytically

There are many uncertainties in the methodology. Statistical population? Sample? Study case? How to collect data? Sampling? Questionnaire questions?

In the discussion, the results of the hypotheses - the reasons for rejecting and accepting the hypotheses and comparing it with previous findings.

The implications are very weak. Rewrite again

6. PLOS authors have the option to publish the peer review history of their article (what does this mean?). If published, this will include your full peer review and any attached files.

Reviewer #1: No

Reviewer #2: No

---

## [Author Response · Author response to Decision Letter 0]

21 Oct 2024

Respons Academic Editor and Reviewer

The red highlighted items in the Revised Manual with Track Changes indicate the modifications made.

Respons Academic Editor:

Response:We submitted the manuscript according to the style requirements of PLOS ONE.

2. Ethical approval

Response: We have prepared an Ethical Review Exemption Confirmation.

Response: We have rechecked the financial disclosures.

4. Please state what role the funders took in the study.

Response:The Outstanding Young Talent Project of Anhui Provincial Department of Education in 2023 (YQYB2023154) provides the direction for this research topic.

Response:The Key Project of Humanities and Social Sciences of Anhui Provincial Department of Education in 2022 (2022AH052746) provides research and interview support for this article.

Response:The Open Fund of Anhui Provincial Key Laboratory of Philosophy and Social Sciences for Intelligent Decision making Development in Copper Industry (2024LXYTYFZXM03) provides labor cost support for this research.

5. We note that you have provided funding information that is not currently declared in your Funding Statement. However, funding information should not appear in the Acknowledgments section or other areas of your manuscript.

Response:We removed the fund information in the Revised Manual with Track Changes and clarified the fund information and its role in the Cover Letter.

6. Please confirm at this time whether or not your submission contains all raw data required to replicate the results of your study. Authors must share the “minimal data set” for their submission. 

Response: We will provide raw data in Excel spreadsheet format.

Response to Reviewer 1 ：

Question: they can improve language and sentence structure.

Response:We appreciate your positive feedback and thorough review of our manuscript, along with your invaluable comments. The author team acknowledges the presence of certain linguistic issues within the text. In response to your feedback, we have conducted another review of the manuscript and engaged a panel of experienced experts to undertake necessary revisions and enhancements. For example 1 : In the abstract section, this article investigates the mechanism of brand communication in short videos and examines strategies for cultivating brand associations to achieve communication goals. Based on signal theory, this study determined that the perceived value of short videos is the source of signals, brand value and transparency are mediating factors, and brand association is the result. This study adopts hypothesis testing and model construction, supplemented by analysis of 560 valid questionnaires on the research platform, to carefully investigate these mechanisms. Simplify , this study investigates how users associate brands in short videos, drawing on signal theory. We explore the mechanisms through which perceived value, brand transparency, and brand value influence brand associations. Using a sample of 560 valid questionnaires. For example 2 : In the introduction, the "2022 China Consumer Trends Report" suggests that brands should utilize multiple online communication channels to enhance their influence, cultivate long-term trust with users, and establish valuable user cognitive assets. It is worth noting that short videos have become an important channel, experiencing rapid growth and attracting a huge user base. The Statistical Report on China's Internet Development (released on March 2, 2023) shows that by the end of 2022, the number of short video users in China will exceed 1 billion, with a utilization rate of 94.8%. Simplify , According to the '2022 China Consumer Trend Report' and the 'China Internet Development Statistics Report', brands should use multiple online communication channels to enhance their influence and build long-term trust with users. Short videos have become a key channel, with over 1 billion users and a usage rate of 94.8%. For example 3 : reorganize and organize research hypotheses, etc.

Response to reviewer 2：

Question 1：In the introduction, no specific scenario is followed. The importance of the subject? Target? Knowledge gap? Innovation? Theoretical and practical contribution?

Response：We appreciate your expert advice. In response to your suggestions, the author team has thoroughly revised the Introduction section, implementing significant changes. 

First, to address the lack of specific scenarios and underscore the study's practical relevance, we have incorporated two successful examples of short video brand marketing: Mattress Firm's "Don't Sleep on Sleep" and Starbucks' "Every Table Has a Story". These instances are juxtaposed with the unsuccessful strategies of Audi and Coconut Tree, highlighting the critical differences. This delineation not only clarifies the essential role of effective brand communication in short videos but also reinforces the importance of the research subject discussed in this paper.

The revised draft is as follows:

In the digital age, the paradigm of brand communication is evolving from traditional one-way dissemination toward more diverse and interactive approaches. The "2022 China Consumer Trend Report" suggests that brands should leverage multiple online communication channels to augment their influence and foster long-term trust with users, thereby building valuable user cognitive assets. Notably, short videos have emerged as a significant channel, experiencing rapid growth and attracting a vast user base. The "Statistical Report on the Development of the Internet in China" (released March 2, 2023) indicates that by the end of 2022, the number of Chinese short video users surpassed one billion, achieving a usage rate of 94.8%(Daily Economic News, 2023) . The varied publishing methods of short videos, coupled with the application of big data technology, have considerably enhanced user reach and provided a more personalized experience. Additionally, the allure of short videos has intensified due to creators' innovative content, making these videos a crucial influence on user decisions and actions (Li and Zhang, 2023) . For instance, "Don’t Sleep on Sleep," a short video by Mattress Firm—the largest mattress retailer in the United States—explores the effects of poor sleep on physical, mental, and emotional health using contextual content, directly boosting sales by 13.7%. Similarly, Starbucks' "Every Table Has a Story" illustrates that the cafe serves as a haven for comfort, inspiration, and community interaction, evidenced by high viewership rates across multiple platforms: 56% on Twitter, 91% on YouTube, and 89% on Facebook/Instagram. Nevertheless, not all brands achieve success through short videos (Zhi hu, 2024) . Audi, for example, suffered reputational damage due to plagiarism in the script of "Moderately Content with Life" (China Business, 2022), and Yeshu Company faced a fine of 400,000 yuan along with criticism from the National Radio and Television Administration for controversial content in their short videos and live broadcasts (Surging news client, 2022) . These challenges underscore the complexities of effective brand communication through short videos, marking it as a critical area of focus.

Second, concerning the knowledge gap you identified, the authors have meticulously reviewed existing research and definitively concluded that previous studies have neglected the impact of brands on product purchase intentions and behaviors. Furthermore, research on brand communication in short video clips is lacking, and scholars have not yet explored the influence of brand characteristics on brand associations in this context. This paper addresses these critical knowledge gaps.

The revised draft is as follows:

Research on brand marketing within short videos remains nascent, primarily concentrating on three core areas: Initially, scholars analyze the attributes of short videos and their influence on brand communication effectiveness. For instance, YAO & KIM (2021) examined authenticity dimensions from a user's perspective, which include originality, relevance, transparency, and experientiality, investigating their impact on both individual and corporate brand communication (Yao and Kim, 2021) . Similarly, Ma and Shao et al. (2020), studying clothing brands, discovered that the characteristics and presentation of short videos directly influence users' perceptions of brand quality(Ma et al., 2020) . Moreover, research suggests that the unique characteristics of users modulate their content receptivity and, consequently, the efficacy of brand marketing. Holmes (2021) explored the correlation between short video users’ self-brand identification and brand communication effectiveness(Holmes, 2021) , while Yang and Zhang et al. (2022) demonstrated that users' self-congruence significantly impacts brand marketing effectiveness in their study of tourism destination brand ambassadors in short videos (Yang et al., 2022) . Additionally, scholars have begun to examine the psychological traits of users during the information reception process as a vital component of their studies. Cho (2020) deemed user experience a crucial factor in analyzing brand awareness in short videos (Cho, 2020) . Aslan Oğuz and Košir et al. (2023) further assessed how user engagement affects brand marketing effectiveness(Aslan Oğuz et al., 2023) . These studies offer fresh insights into the integration of brand information within short videos and delve deeply into brand-related intentions and behaviors. However, the crucial initial step of brand recall by short video users for effective marketing has yet to be addressed in current research. Additionally, while it is evident that a brand's inherent characteristics impact its marketing success in short videos, this area remains unexplored.

Third, addressing the issues of 'Target' and 'Innovation' you highlighted, the authors have revised the article’s third paragraph. The opening sentence succinctly outlines the research's main objectives, grounded in practical needs and identified theoretical gaps. The theoretical contributions are explicitly stated in the fourth paragraph. Regarding practical contributions, although briefly mentioned in the context of managerial implications in the third paragraph, they are elaborated in detail in the article's sixth section due to space limitations.

The revised draft is as follows:

This article, grounded in the framework of signal theory, examines effective strategies for disseminating brand messaging within short video environments and delineates a mechanism that enhances corporate brand marketing efficiency and effectiveness. The study addresses three principal questions: first, what elements in short videos enable users to perceive brand information; second, which brand features influence associations in these videos; and third, what mechanisms underlie these brand associations. Initially employing signal theory, the research investigates the perceived values users extract from short videos and their specific contributing factors. Data gathered through online surveys subsequently allows for an empirical analysis, which confirms the relationship between these factors and brand associations. Brand transparency and value are then examined as mediating variables, their roles verified to explore how they influence the formation of associations in short videos. Finally, informed by these findings, the paper offers actionable recommendations for enterprises on brand dissemination strategies in short videos.

The theoretical contribution of this paper lies in its exploration and validation of critical factors in the dissemination of brand information in short videos through signal theory: perceived values, brand features (transparency and value), and user brand associations. Moreover, the study elucidates the mechanisms of these factors, thereby broadening the scope of research in brand marketing within short video contexts and extending the application reach of signal theory.

Question 2：In the literature, it has been tried to make an essay rather than analyzing the literature and examining the relationships between variables analytically

Response：You noted that the literature review section was rather disorganized due to the merging of 'Theoretical Background' and 'Hypotheses' into a single section, which was improper. After careful consideration, the paper will now divide these into 'Theoretical Background' and 'Hypotheses Development' sections to prevent confusion. Regarding the research review, the current paper has already structured the existing studies while describing the theoretical background in the second paragraph. To maintain the paper's logical coherence, it will not include a separate literature review section.

Question3: There are many uncertainties in the methodology. Statistical population? Sample? Study case? How to collect data? Sampling? Questionnaire questions?

Response: We appreciate your identification of the oversight. To address this, we have implemented the following corrections: 

First,a new section titled 'Sample and Study Case' has been added to clarify the selection of Chinese users as the research subjects for this study. This section also details the methodical distribution and randomness of the study sample via an online survey platform.

The revised draft is as follows:

3.3 Sample and Study Case

The China Online Audio-visual Development Report (2024) reveals that by December 2023, the number of online audio-visual users in China reached 1.074 billion, with a total of 1.55 billion short video accounts across the internet (China Online Audio-Visual Association, 2024). As a result, there is substantial usage and deep market penetration of short videos among Chinese users (Tian et al., 2023)). Considering China's vast population engaged in using short videos, alongside rapid economic growth and the pervasive availability of smart devices, the conditions for usage are favorable and experiences are extensive. Thus, this demographic has been selected as the sample for this study. To guarantee the randomness of the research, surveys will be collected randomly using the paid questionnaire platform Credamo, with participants

Second, from the inception of the questionnaire design, this study meticulously considered both its professionalism and readability. Initially, expert colleagues were invited to assess the draft questionnaire. Furthermore, a preliminary survey was conducted to gather feedback on the wording of the items, ensuring the comprehensibility of the final version. This revision has been relocated to the second paragraph of Section 4.2, "Questionnaire Design."

The revised draft is as follows:

Upon drafting the initial survey questionnaire, the authors sought to enhance both its professionalism and relevance. They engaged peer reviewers to critique the questionnaire, implementing adjustments based on this feedback. Furthermore, to ensure comprehensive understanding of the survey questions among respondents, a preliminary survey was conducted. Comments from this early feedback prompted refinements in the question wording, leading to the finalization of the official research questionnaire.

Third, in response to feedback, the newly expanded "Data Collection" section now comprehensively outlines the rationale behind the selection of the data platform for the survey. It also details the preliminary criteria used for screening the data, as well as the quantity and ratio of valid questionnaires collected.

The revised draft is as follows:

4.4 Data Collection

This study utilized Credamo, a leading paid survey platform in China known for its comprehensive services including questionnaire design, access to millions of online participants, and advanced visual statistical modeling. Credamo, an acronym for Creator of Data and Model (https://www.credamo.com/), serves over 3,000 universities worldwide and 4,000 companies, accommodating over 3 million online participants. Its robust dataset facilitates the identification of participants whose profiles align pr

---

## [Decision Letter · Decision Letter 1]

19 Dec 2024

How are brands associated by users in short videos - A study on the mechanism of user associations with brand placements in short videos based on signal theory

PONE-D-24-22415R1

Dear Dr. Sun,

We’re pleased to inform you that your manuscript has been judged scientifically suitable for publication and will be formally accepted for publication once it meets all outstanding technical requirements.

Kind regards,

Reza Rostamzadeh

Academic Editor

PLOS ONE

Additional Editor Comments (optional):

Reviewers' comments:

Reviewer's Responses to Questions

**Comments to the Author**

1. If the authors have adequately addressed your comments raised in a previous round of review and you feel that this manuscript is now acceptable for publication, you may indicate that here to bypass the “Comments to the Author” section, enter your conflict of interest statement in the “Confidential to Editor” section, and submit your "Accept" recommendation.

Reviewer #2: All comments have been addressed

2. Is the manuscript technically sound, and do the data support the conclusions?

Reviewer #2: Yes

3. Has the statistical analysis been performed appropriately and rigorously? 

Reviewer #2: Yes

4. Have the authors made all data underlying the findings in their manuscript fully available?

Reviewer #2: Yes

5. Is the manuscript presented in an intelligible fashion and written in standard English?

Reviewer #2: Yes

6. Review Comments to the Author

Reviewer #2: Thank you for the opportunity to review the manuscript.

According to the review done by the authors, the manuscript is acceptable. Congratulations to the authors

7. PLOS authors have the option to publish the peer review history of their article (what does this mean?). If published, this will include your full peer review and any attached files.

Reviewer #2: No

---

## [Editor Report · Acceptance letter]

28 Dec 2024

PONE-D-24-22415R1 

PLOS ONE

Dear Dr. Dongqi, 

I'm pleased to inform you that your manuscript has been deemed suitable for publication in PLOS ONE. Congratulations! Your manuscript is now being handed over to our production team.

Kind regards, 

on behalf of

Dr. Reza Rostamzadeh 

Academic Editor

PLOS ONE